# Enhancing the Transferability of Adversarial Examples with Noise Reduced Gradient

## Abstract

Deep neural networks provide state-of-the-art performance for many applications of interest. Unfortunately they are known to be vulnerable to adversarial examples, formed by applying small but malicious perturbations to the original inputs. Moreover, the perturbations can *transfer across models*: adversarial examples generated for a specific model will often mislead other unseen models. Consequently the adversary can leverage it to attack against the deployed black-box systems.

In this work, we demonstrate that the adversarial perturbation can be decomposed into two components: model-specific and data-dependent one, and it is the latter that mainly contributes to the transferability. Motivated by this understanding, we propose to craft adversarial examples by utilizing the noise reduced gradient (NRG) which approximates the data-dependent component. Experiments on various classification models trained on ImageNet demonstrates that the new approach enhances the transferability dramatically. We also find that low-capacity models have more powerful attack capability than high-capacity counterparts, under the condition that they have comparable test performance. These insights give rise to a principled manner to construct adversarial examples with high success rates and could potentially provide us guidance for designing effective defense approaches against black-box attacks.

## 1 Introduction

With the resurgence of neural networks, more and more large neural network models are applied in real-world applications, such as speech recognition, computer vision, etc. While these models have exhibited good performance, recent works (Szegedy et al. (2013); Goodfellow et al. (2014)) show that an adversary is always able to fool the model into producing incorrect outputs by manipulating the inputs maliciously. The corresponding manipulated samples are called *adversarial examples*. However, how to understand this phenomenon (Goodfellow et al. (2014); Tramèr et al. (2017b)) and how to defend against adversarial examples effectively (Kurakin et al. (2016); Tramèr et al. (2017a); Carlini & Wagner (2017)) are still open questions. Meanwhile it is found that adversarial examples can transfer across different models, i.e., the adversarial examples generated from one model can also fool another model with a high probability. We refer to such property as *transferability*, which can be leveraged to attack black-box systems (Papernot et al. (2016); Liu et al. (2016)).

The phenomenon of adversarial vulnerability was first introduced and studied in Szegedy et al. (2013). The authors modeled the adversarial example generation as an optimization problem solved by box-constraint L-BFGS, and also attributed the presence of adversarial examples to the strong nonlinearity of deep neural networks. Goodfellow et al. (2014) argued instead that the primary cause of the adversarial instability is the linear nature and the high dimensionality, and the view yielded the fast gradient sign method (FGSM). Similarly based on an iterative linearization of the classifier, Moosavi-Dezfooli et al. (2015) proposed the DeepFool method. In Kurakin et al. (2016); Tramèr et al. (2017a), it was shown that the iterative gradient sign method provides stronger white-box attacks but does not work well for black-box attacks. Liu et al. (2016) analyzed the transferability of adversarial examples in detail and proposed *ensemble-based approaches* for effective black-box attacks. In Carlini & Wagner (2017) it was demonstrated that high-confidence adversarial examples that are strongly misclassified by the original model have stronger transferability.

In addition to crafting adversarial examples for attacks, there exist lots of works on devising more effective defense. Papernot et al. (2015) proposed the defensive distillation. Goodfellow et al. (2014) introduced the adversarial training method, which was examined on ImageNet by Kurakin et al. (2016) and Tramèr et al. (2017a). Lu et al. (2017) utilized image transformation, such as rotation, translation, and scaling, etc, to alleviate the harm of the adversarial perturbation. Instead of making the classifier itself more robust, several works (Li & Li (2016); Feinman et al. (2017)) attempted to detect the adversarial examples, followed by certain manual processing. Unfortunately, all of them can be easily broken by designing stronger and more robust adversarial examples (Carlini & Wagner (2017); Athalye & Sutskever (2017)).

In this work, we give an explanation for the transferability of adversarial examples and use the insight to enhance black-box attacks. Our key observation is that adversarial perturbation can be decomposed into two components: model-specific and data-dependent one. The model-specific component comes from the model architecture and random initialization, which is noisy and represents the behavior off the data manifold. In contrast, the data-dependent component is smooth and approximates the ground truth on the data manifold. We argue that it is the data-dependent part that mainly contributes to the transferability of adversarial perturbations across different models. Based on this view, we propose to construct adversarial examples by employing the data-dependent component of gradient instead of the gradient itself. Since this component is estimated via noise reduction strategy, we call it *noise-reduced gradient (NRG)* method. Benchmark on the ImageNet validation set demonstrates that the proposed noise reduced gradient used in conjunction with other known methods could dramatically increase the success rate of black-box attacks. to perform black-box attacks over ImageNet validation set.

We also explore the dependence of success rate of black-box attacks on model-specific factors, such as model capacity and accuracy. We demonstrate that models with higher accuracy and lower capacity show stronger capability to attack unseen models. Moreover this phenomenon can be explained by our understanding of transferability, and may provide us some guidances to attack unseen models.

## 2 BACKGROUND

### 2.1 ADVERSARIAL EXAMPLES

We use $f : \mathbb{R}^d \mapsto \mathbb{R}^K$ to denote the model function, which is obtained via minimizing the empirical risk over training set. For simplicity we omit the dependence on the trainable model parameter $\boldsymbol{\theta}$, since it is fixed in this paper. For many applications of interest, we always have $d \gg 1$ and $K = o(1)$. According to the local linear analysis in Goodfellow et al. (2014), the high dimensionality makes $f(\boldsymbol{x})$ inevitably vulnerable to the adversarial perturbation. That is, for each $\boldsymbol{x}$, there exists a small perturbation $\boldsymbol{\eta}$ that is nearly imperceptible to human eyes, such that

$$f(\boldsymbol{x}) = y^{\text{true}}, \quad f(\boldsymbol{x} + \boldsymbol{\eta}) \neq y^{\text{true}} \tag{1}$$

We call $\boldsymbol{\eta}$ adversarial perturbation and correspondingly $x^{adv} := \boldsymbol{x} + \boldsymbol{\eta}$ adversarial example. In this work, we mainly study the adversarial examples in the context of deep neural networks, though they also exist in other models, for example, support vector machine (SVM) and decision tree, etc (Papernot et al. (2016)).

We call the attack (1) a non-targeted attack because the adversary has no control of the output other than requiring $\boldsymbol{x}$ to be misclassified by the model.

In contrast, a *targeted attack* aims at fooling the model into producing a wrong label specified by the adversary. I.e.

$$f(\boldsymbol{x} + \boldsymbol{\eta}) = y^{\text{target}}.$$

In contrast, we call the attack (1) a non-targeted attack.

In the *black-box attack* setting, the adversary has no knowledge of the target model (e.g. architecture and parameters) and is not allowed to query the model. That is, the target model is a pure black-box. However the adversary can construct adversarial examples on a local model (also called the source model) that is trained on the same or similar dataset with the target model. Then it deploys those adversarial examples to fool the target model. This is typically referred to as a black-box attack, as opposed to the white-box attack whose target is the source model itself.

## 2.2 MODELLING OF GENERATING ADVERSARIAL EXAMPLES

In general, crafting adversarial perturbation can be modeled as following optimization problem,

$$\text{maximize} \quad J(f(\boldsymbol{x} + \boldsymbol{\eta}), y^{\text{true}})$$
$$\text{s.t.} \quad \|\boldsymbol{\eta}\| \leq \varepsilon, \tag{2}$$

where $J$ is a loss function measuring the discrepancy between the prediction and ground truth; $\| \cdot \|$ is the metric to quantify the magnitude of the perturbation. For image data, there is also an implicit constraint: $\boldsymbol{x}^{adv} \in [0, 255]^d$, with $d$ being the number of pixels. In practice, the commonly choice of $J$ is the *cross entropy*. Carlini & Wagner (2016) introduced a loss function that directly manipulates the output logit instead of probability. This loss function is also adopted by many works. As for the measurement of distortion, The best metric should be human eyes, which is unfortunately difficult to quantify. In practice, $\ell_\infty$ and $\ell_2$ norms are commonly used.

**Ensemble-based approaches**    To improve the strength of adversarial examples, instead of using a single model, Liu et al. (2016) suggest using a large ensemble consisting of $f_1, f_2, \cdots, f_Q$ as our source models. Although there exist several ensemble strategies, similar as Liu et al. (2016), we only consider the most commonly used method, averaging the predicted probability of each model. The corresponding objective can be written as

$$\text{maximize} \quad J\left(\sum_{i=1}^{Q} w_i f_i(\boldsymbol{x} + \boldsymbol{\eta}), y^{\text{true}}\right)$$
$$\text{s.t.} \quad \|\boldsymbol{\eta}\| \leq \varepsilon \tag{3}$$

where $w_i$ are the ensemble weights with $\sum w_i = 1$.

Objectives (2) and (3) are for non-targeted attacks, and the targeted counterpart can be derived similarly.

## 2.3 OPTIMIZER

There are various optimizers to solve problem (2) and (3). In this paper, we mainly use the normalized-gradient based optimizer.

**Fast Gradient Based Method**    This method (Goodfellow et al. (2014)) attempts to solve (2) by performing only one step iteration,

$$\boldsymbol{x}^{adv} \leftarrow \boldsymbol{x} + \varepsilon\, g(\boldsymbol{x}), \tag{4}$$

where $g(\boldsymbol{x})$ is a normalized gradient vector. For $\ell_\infty$-attack ,

$$g^\infty(\boldsymbol{x}) = \text{sign}\left(\nabla_x J(f(\boldsymbol{x}); y^{\text{true}})\right) \quad \textbf{(FGSM)}; \tag{5}$$

similarly for $\ell_q$-attack,

$$g^q(\boldsymbol{x}) = \frac{\nabla_x J(f(\boldsymbol{x}); y^{\text{true}})}{\|\nabla_x J(f(\boldsymbol{x}); y^{\text{true}})\|_q} \quad \textbf{(FGM)}. \tag{6}$$

Both of them are called fast gradient based method (FGBM) and are empirically shown to be fast. Also they have very good transferability (Kurakin et al. (2016); Tramèr et al. (2017a)) though not optimal. So it is worth considering this simple yet effective optimizer.

**Iterative Gradient Method**    This method (Madry et al. (2017); Kurakin et al. (2016)) performs the projected normalized-gradient ascent

$$\boldsymbol{x}^{t+1} \leftarrow \text{clip}^{\boldsymbol{x}^0, \varepsilon}\left(\boldsymbol{x}^t + \alpha\, g(\boldsymbol{x}^t)\right) \tag{7}$$

for $k$ steps, where $\boldsymbol{x}^0$ is the original clean image; $\text{clip}^{\boldsymbol{x}, \varepsilon}(\cdot)$ is the projection operator to enforce $\boldsymbol{x}^t \in [0, 255]^d \cap \{\boldsymbol{x} \mid \|\boldsymbol{x} - \boldsymbol{x}^0\| \leq \varepsilon\}$

and $\alpha$ is the step size. In analogous to the fast gradient based method, the normalized gradient $g(\boldsymbol{x})$ is chosen as $g^\infty(\boldsymbol{x})$ for $\ell_\infty$-attack, called iterative gradient sign method (IGSM), and $g^q(\boldsymbol{x})$ for $\ell_q$-attack. The fast gradient based method (4) can be viewed as a special case of (7) when $\alpha = \varepsilon, k = 1$.

# 3 ENHANCING THE TRANSFERABILITY OF ADVERSARIAL EXAMPLES

## 3.1 DATA-DEPENDENT COMPONENTS ENABLE TRANSFERABILITY

There are few articles trying to understand why adversarial examples can transfer between models, though it is extremely important for performing black-box attacks and building successful defenses. To the best of our knowledge, the only two works are (Liu et al. (2016); Tramèr et al. (2017b)), which suggested that the transferability comes from the similarity between the decision boundaries of the source and target models, especially in the direction of transferable adversarial examples. Tramèr et al. (2017b) also claimed that transferable adversarial examples span a contiguous subspace.

To investigate the transfer phenomenon, we ask a further question: *what similarity between the different models A and B that enables transferability of adversarial examples across them?* Since the model $A$ and $B$ have a high performance on the same dataset, they must have learned a similar function *on the data manifold*. However, the behaviour of the models *off the data manifold* can be different. This is determined by the architectures of the models and random initializations, both of which are data-independent factors.

This clearly hints us to decompose the perturbation into two factors on and off the data manifold. We referred to them as *data-dependent* and *model-specific* components. We hypothesize that the component on the data manifold mainly contributes to the transferability from $A$ to $B$, since this component captures the data-dependent information shared between models. The model-specific one contributes little to the transferability due to its different behaviours off the data manifold for different models.

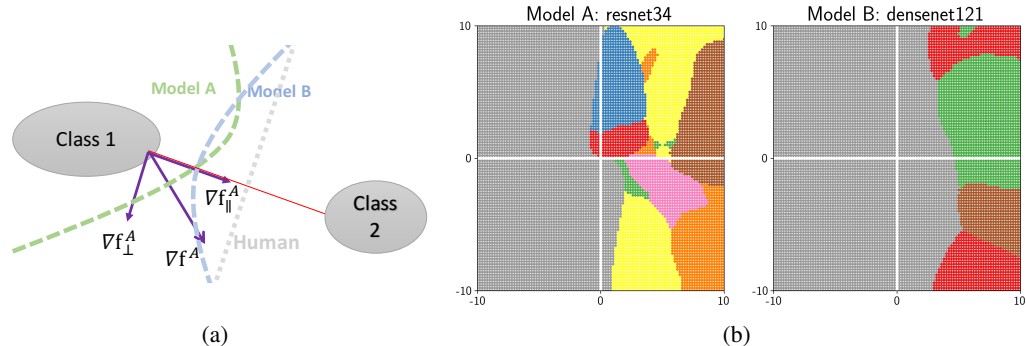

(a)                                               (b)

Figure 1: (a) Illustration of the decomposition and transferability of the adversarial perturbation $\nabla f^A$. (b) Visualization of decision boundaries. Each point represents an image: $\boldsymbol{x} + u \nabla f_{\parallel}^A / \|\nabla f_{\parallel}^A\|_2 + v \nabla f_{\perp}^A / \|\nabla f_{\perp}^A\|_2$, and the same color indicates the same predicted label. More details can be found in Section 6.3.

Take $\boldsymbol{\eta} = \nabla f^A$ as the adversarial perturbation crafted from model $A$, we illustrate this explanation in Figure 1. In the left panel, the decision boundaries of two models are similar in the inter-class area. As can be observed, $\nabla f^A$ can mislead both model $A$ and $B$. Then we decompose the perturbation into two parts, a data-dependent component $\nabla f_{\parallel}^A$ and a model-specific one $\nabla f_{\perp}^A$, respectively. Since $\nabla f_{\parallel}^A$ is almost aligned to the inter-class deviation (red line), i.e. on the data manifold, it can attack model $B$ easily. However, The model-specific $\nabla f_{\perp}^A$ contributes little to the transfer from $A$ to $B$, though it can successfully fool model $A$ with a very small distortion.

In the right panel, we plot the decision boundaries of *resnet34* (model $A$) and *densenet121* (model $B$) for ImageNet. The horizontal axis represents the direction of data-dependent component $\nabla f_{\parallel}^A$ estimated by our proposed NRG method (8) from *resnet34*; and the vertical axis depicts the direction of model-specific component $\nabla f_{\perp}^A$. It is easily observed that for model $A$ *resnet34*, a small perturbation in both directions can produce wrong classification. However, when applied to model $B$ *densenet121*, a large perturbation along the $\nabla f_{\perp}^A$ direction cannot change the classification results, while a small perturbation along the data-dependent one can change the prediction easily.

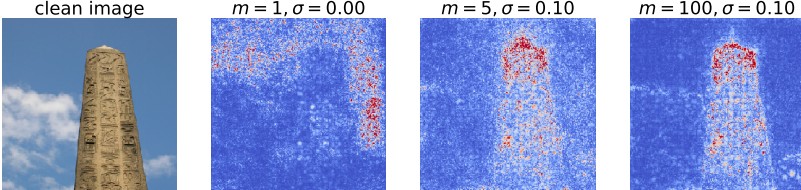

Figure 2: Visualization of the gradient with respect to input for *resnet152*. The leftmost is the original image, and the rest three images are the visualization of the gradients.

This understanding suggests that to increase success rates of black-box adversarial attacks, we should enhance the data-dependent component. We hence propose the NRG method which achieves that by reducing the model-specific noise, as elaborated in the following.

### 3.2 NOISE-REDUCED GRADIENT METHODS

**Noise-reduced gradient (NRG)** The model-specific component $\nabla f_\perp$ inherits from the random initialization, so it must be very noisy, i.e. high-frequency (the same observation is also systematically investigated in Balduzzi et al. (2017)); while $\nabla f_\parallel$ is smooth, i.e. low-frequency, since it encodes the knowledge learned from training data. Therefore, *local average* can be applied to remove the noisy model-specific information, yielding an approximation of the data-dependent component,

$$\nabla f_\parallel \approx \frac{1}{m}\sum_{i=1}^{m} \nabla f(\boldsymbol{x}+\boldsymbol{\xi}_i), \quad \boldsymbol{\xi}_i \sim \mathcal{N}(0,\sigma^2 I_d), \tag{8}$$

where the $m$ is the number of samples chosen for averaging. We call (8) *noise-reduced gradient* (NRG), which captures more data-dependent information than ordinary gradient $\nabla f$. To justify the effectiveness of this method, similar to Smilkov et al. (2017), we visualize NRG for various $m$ in Figure 2. As shown, larger $m$ leads to a smoother and more data-dependent gradient, especially for $m = 100$ the NRG can capture the semantic information of the obelisk.

The noisy model-specific information of $\nabla f$ could mislead the optimizer into the solutions that are overfitting to the specific source model, thus generalizing poorly to the other model. Therefore we propose to perform attacks by using $\nabla f_\parallel$ in Eq.(8), instead of $\nabla f$, which can drive the optimizer towards the solutions that are more data-dependent.

**Noise-reduced Iterative Sign Gradient Method (nr-IGSM)** The iterative gradient sign method mounted by NRG can be written as,

$$G^t = \frac{1}{m}\sum_{i=1}^{m} \nabla J(\boldsymbol{x}^t+\boldsymbol{\xi}_i), \quad \boldsymbol{\xi}_i \sim \mathcal{N}(0,\sigma^2 I_d)$$
$$\boldsymbol{x}^{t+1} = \text{clip}^{\boldsymbol{x}^0,\varepsilon}\left(\boldsymbol{x}^t + \alpha\,\text{sign}\left(G^t\right)\right), \tag{9}$$

The special case $k = 1, \alpha = \varepsilon$ is called noise-reduced fast gradient sign method (nr-FGSM) accordingly. For $\ell_q$-attack, the noise-reduced version is similar, replacing the second equation of (9) by

$$\boldsymbol{x}^{t+1} = \text{clip}^{\boldsymbol{x}^0,\varepsilon}\left(\boldsymbol{x}^t + \alpha\,G^t/\|G^t\|_q\right). \tag{10}$$

For any general optimizer, the corresponding noise-reduced counterpart can be derived similarly.

## 4 EXPERIMENT SETTING

To justify and analyze the effectiveness of NRG for enhancing the transferability, we use the start-of-the-art classification models trained on ImageNet dataset. We elaborate the details in the following.

**Dataset** We use the ImageNet ILSVRC2012 validation set that contains $50,000$ samples. For each attack experiment, we randomly select $5,000$ images that can be correctly recognized by all the

models, since it is meaningless to construct adversarial perturbations for the images that target models cannot classify correctly. And for the targeted attack experiments, each image is specified by a random wrong label.

**Models** We use the pre-trained models provided by PyTorch including *resnet18, resnet34, resnet50, resnet101, resnet152, vgg11_bn, vgg13_bn, vgg16_bn, vgg19_bn, densenet121, densenet161, densenet169, densenet201, alexnet, squeezenet1_1*. The Top-1 and Top-5 accuracies of them can be found on website[1]. To increas the reliability of experiments, all the models have been used, however for a specific experiment we only choose several of them to save computational time.

**Criteria** Given a set of adversarial examples, $\{(\boldsymbol{x}_1^{adv}, y_1^{\text{true}}), (\boldsymbol{x}_2^{adv}, y_2^{\text{true}}), \ldots, (\boldsymbol{x}_N^{adv}, y_N^{\text{true}})\}$, We calculate their *Top-1 success rates* fooling a given model $F(\boldsymbol{x})$ by

$$\frac{1}{N} \sum_{i=1}^{N} 1[F(\boldsymbol{x}_i^{adv}) \neq y_i^{\text{true}}]. \tag{11}$$

If $F$ is the model used to generate adversarial examples, then the rate indicates the the white-box attack performance. For targeted attacks, each image $\boldsymbol{x}^{adv}$ is associated with a pre-specified label $y^{\text{target}} \neq y^{\text{true}}$. Then we evaluate the performance of the targeted attack by the following *Top-1 success rate*,

$$\frac{1}{N} \sum_{i=1}^{N} 1[F(\boldsymbol{x}_i^{adv}) = y_i^{\text{target}}]. \tag{12}$$

The corresponding Top-5 rates can be computed in a similar way.

**Attacks** Throughout this paper the cross entropy [2] is chosen as our loss function $J$. We measure the distortion by two distances: $\ell_\infty$ norm and scaled $\ell_2$ norm, i.e. *root mean square deviation* (RMSD) $\sqrt{\sum_{i=1}^{d} \eta_i^2 / d}$, where $d$ is the dimensionality of inputs. As for optimizers, both FGSM and IGSM are considered.

## 5 EFFECTIVENESS OF NOISE-REDUCED METHODS

In this section we demonstrate the effectiveness of our noise-reduced gradient technique by combining it with several commonly-used methods.

### 5.1 SINGLE-MODEL BASED APPROACHES

**Fast gradient based method** We first examine the combination of noise reduced gradient and fast gradient based methods. The success rates of FGSM and nr-FGSM are summarized in Table 1 (results of FGM $\ell_2$-attacks can be found in Appendix C). We observe that, for any pair of black-box attack, nr-FGSM performs better than original FGSM consistently and dramatically. Even the white-box attacks (the diagonal cells) also have improvements. This result implies that the direction of noise-reduced gradient is indeed more effective than the vanilla gradient for enhancing the transferability.

**Iterative gradient sign method** One may argue that the above comparison is unfair, since nr-FGSM consumes more computational cost than FGSM, determined by $m$: number of perturbed inputs used for local average. Here we examine IGSM and nr-IGSM under the same number of gradient calculations. Table 2 presents the results. Except for the attacks from *alexnet*, as we expect, adversarial examples generated by nr-IGSM indeed transfer much more easily than those generated by IGSM. This indicates that the noise-reduced gradient (NRG) does guide the optimizer to explore the more data-dependent solutions.

**Some observations** By comparing Table 1 with Table 2, we find that large models are more robust to adversarial transfer than small models, for example the *resnet152*. This phenomenon has also been extensively investigated by Madry et al. (2017). It also shows that the transfer among the

---

[1]http://pytorch.org/docs/master/torchvision/models.html
[2]We also tried the loss described in Carlini & Wagner (2016) but did not find its superiority to cross entropy. We guess the reason is that hard constraints instead of soft penalizations are used in our formulation.

Table 1: Top-1 success rates of non-targeted FGSM and nr-FGSM attacks between pairs of models. The cell $(S, T)$ indicates the percentage of adversarial examples generated for model $S$ (row) that successfully fool model $T$ (column). For each cell, the left is the success rate of FGSM; the right is the one of nr-FGSM. In this experiment, distortion $\varepsilon = 15$.

|  | alexnet | densenet121 | resnet152 | resnet34 | vgg13_bn | vgg19_bn |
|---|---|---|---|---|---|---|
| alexnet | 98.3 / 98.1 | 24.7 / 46.0 | 18.0 / 19.3 | 34.3 / 37.8 | 48.6 / 50.2 | 33.7 / 43.7 |
| densenet121 | 58.2 / 62.1 | 91.2 / 98.8 | 34.4 / **66.7** | 46.2 / **74.5** | 53.0 / **72.5** | 44.9 /**71.4** |
| resnet152 | 53.2 / 56.9 | 39.2 / 67.2 | 81.4 / 95.9 | 45.4 / 71.3 | 43.3 / 62.4 | 36.8 / 61.5 |
| resnet34 | 57.3 / **62.5** | 46.3 / **71.1** | 38.4 / 66.4 | 94.3 / 98.5 | 54.4 / 70.5 | 46.7 / 68.8 |
| vgg13_bn | 46.8 / 53.9 | 23.0 / 48.4 | 16.0 / 34.4 | 28.0 / 53.2 | 96.7 / 98.7 | 54.2 / 84.6 |
| vgg19_bn | 47.4 / 54.6 | 28.1 / 58.8 | 18.7 / 46.4 | 31.5 / 62.3 | 62.2 / 87.1 | 91.1 / 98.3 |

Table 2: Top-1 success rates of non-targeted IGSM and nr-IGSM attacks between pairs of models. The cell $(S,T)$ indicates the percentage of adversarial examples generated for model $S$ (row) that successfully fool model $T$ (column). For each cell: (1) the left is the success rate of IGSM ($k = 100, \alpha = 1$); (2) the right is the one of nr-IGSM ($m = 20, \sigma = 15, k = 5, \alpha = 5$). In this experiment, distortion $\varepsilon = 15$.

|  | alexnet | densenet121 | resnet152 | resnet34 | vgg13_bn | vgg19_bn |
|---|---|---|---|---|---|---|
| alexnet | 100 / 100 | 26.9 / 24.6 | 18.3 / 16.0 | 38.6 / 37.1 | 49.2 / 47.3 | 35.9 / 34.4 |
| densenet121 | 30.6 / **46.8** | 100 / 99.9 | 50.1 / **80.6** | 59.9 / **87.2** | 62.2 / **82.2** | 56.5 / **84.3** |
| resnet152 | 27.4 / 40.7 | **52.5 / 81.3** | 100 / 100 | 57.2 / 85.6 | 47.7 / 71.1 | 42.9 / 72.6 |
| resnet34 | 29.7 / 44.5 | 51.5 / 76.4 | 46.5 / 73.1 | 100 / 100 | 53.8 / 74.8 | 49.1 / 74.5 |
| vgg13_bn | 28.4 / 41.2 | 24.1 / 49.2 | 14.3 / 33.5 | 25.1 / 54.1 | 100 / 100 | 90.6 / **96.4** |
| vgg19_bn | 24.9 / 39.2 | 27.1 / 57.5 | 16.7 / 41.6 | 27.6 / 60.7 | 92.0 / **96.1** | 99.9 / 99.9 |

models with similar architectures is much easier, such as the $90\%+$ transferability between vgg-style networks in Table 2. This implies that the model-specific component also contributes to the transfer across models with similar architectures.

Additionally in most cases, IGSM generates stronger adversarial examples than FGSM except the attacks against *alexnet*. This contradicts the claims in Kurakin et al. (2016) and Tramèr et al. (2017a) that adversarial examples generated by FGSM transfer more easily than the ones of IGSM. However our observation is consistent with the conclusions of Carlini & Wagner (2016): the higher confidence (smaller cross entropy) adversarial examples have in the source model, the more likely they will transfer to the target model. We conjecture that this is due to the inappropriate choice of hyperparameters, for instance $\alpha = 1, k = \min(\varepsilon + 4, 1.24\varepsilon)$ in Kurakin et al. (2016) are too small, and the solution has not fit the source model enough (i.e. underfitting). When treated as a target model to be attacked, the *alexnet* is significantly different from those source models in terms of both architecture and test accuracy. And therefore, the multiple iterations cause the IGSM to overfit more than FGSM, producing a lower fooling rate.

These phenomena clearly indicate that we should not trust the objective in Eq. (2) completely, which might cause the solution to overfit to the source model-specific information and leads to poor transferability. Our noise reduced gradient technique regularizes the optimizer by removing the model-specific information from the original gradients, and consequently, it can converge to a more data-dependent solution that has much better cross-model generalization capability.

## 5.2 ENSEMBLE BASED APPROACHES

In this part, we apply NRG method into ensemble-based approaches described in Eq. (3). Due to the high computational cost of model ensembles, we select $1,000$ images, instead of $5,000$, as our evaluation set.

For non-targeted attacks, both FGSM, IGSM and their noise reduced versions are tested. The Top-1 success rates of IGSM attacks are nearly saturated, so we report the corresponding Top-5 rates in Table 3(a) to demonstrate the improvements of our methods more clearly. The results of FGSM and nr-FGSM attacks can be found in Appendix C.

For targeted attacks, to generate an adversarial example predicted by unseen target models as a specific label, is much harder than generating non-targeted examples. Liu et al. (2016) demonstrated that single-model based approaches are very ineffective in generating targeted adversarial examples. That is why we did not consider targeted attacks in Section 5.1.

Different from the non-targeted attacks, we find that targeted adversarial examples are sensitive to the step size $\alpha$ used in the optimization procedures (6) and (9). After trying lots of $\alpha$, we find that **a large step size is necessary for generating strong targeted adversarial examples**. Readers can refer to Appendix A for more detailed analysis on this issue, though we cannot fully understand it. Therefore we use a much larger step size compared to the non-target attacks. The Top-5 success rates are reported in Table 3(b).

By comparing success rates of normal methods and our proposed NRG methods in Table 3 for both targeted and non-targeted attacks, we observed that NRG methods outperform the corresponding normal methods by a remarkable large margin in this scenario.

Table 3: Top-5 success rates of ensemble-based approaches. The cell $(S, T)$ indicates the attack performances from the ensemble $S$ (row) against the target model $T$ (column). For each cell: the left is the rate of normal method, in contrast the right is the one of the noise-reduced counterpart. The corresponding Top-1 success rates can be found in Appendix C (Table 7 and Table 8).

(a) **Non-targeted attacks**: IGSM ($k = 200, \alpha = 1$) versus nr-IGSM ($k = 10, \alpha = 5, m = 20, \sigma = 15$), distortion $\varepsilon = 15$.

| Ensemble | densenet121 | resnet152 | resnet50 | vgg13_bn |
|---|---|---|---|---|
| resnet18+resnet34+resnet101 | 43.0 / **75.5** | 54.5 / **81.6** | 62.6 / **85.4** | 42.0 / 74.2 |
| vgg11_bn +densenet161 | 40.5 / 73.5 | 18.5 / 56.4 | 33.4 / 70.2 | 68.3 / 85.6 |
| resnet34+vgg16_bn+alexnet | 26.5 / 65.2 | 15.7 / 55.3 | 33.8 / 72.8 | 77.8 / **89.9** |

(b) **Targeted attacks**: IGSM ($k = 20, \alpha = 15$) versus nr-IGSM ($k = 20, \alpha = 15, m = 20, \sigma = 15$), distortion $\varepsilon = 20$.

| Ensemble | resnet152 | resnet50 | vgg13_bn | vgg16_bn |
|---|---|---|---|---|
| resnet101+densenet121 | 28.1 / 56.8 | 26.2 / 52.4 | 7.7 / 23.6 | 8.1 / 29.7 |
| resnet18+resnet34+resnet101+densenet121 | 50.4 / **70.4** | 54.7 / **72.4** | 23.2 / **44.2** | 28.1 / 52.6 |
| vgg11_bn+vgg13_bn+resnet18 +resnet34+densenet121 | 24.3 / 55.8 | 36.9 / 65.9 | - | 62.2 / **83.5** |

# 6 ANALYSIS

## 6.1 INFLUENCE OF HYPER PARAMETERS

In this part, we explore the sensitivity of hyper parameters $m, \sigma$ when applying our NRG methods for black-box attacks. We take nr-FGSM approach as a testbed over the selected evaluation set described in Section 4. Four attacks are considered here, and the results are shown in Figure 3. It is not surprising that larger $m$ leads to higher fooling rate for any distortion level $\varepsilon$ due to the better estimation of the data-dependent direction of the gradient. We find there is an optimal value of $\sigma$ inducing the best performance. Overly large $\sigma$ will introduce a large bias in (8). Extremely small $\sigma$ is unable to remove the noisy model-specific information effectively, since noisy components of gradients of different perturbed inputs are still strongly correlated for small $\sigma$. Moreover the optimal $\sigma$ varies for different source models, and in this experiment it is about 15 for *resnet18*, compared to 20 for *densenet161*.

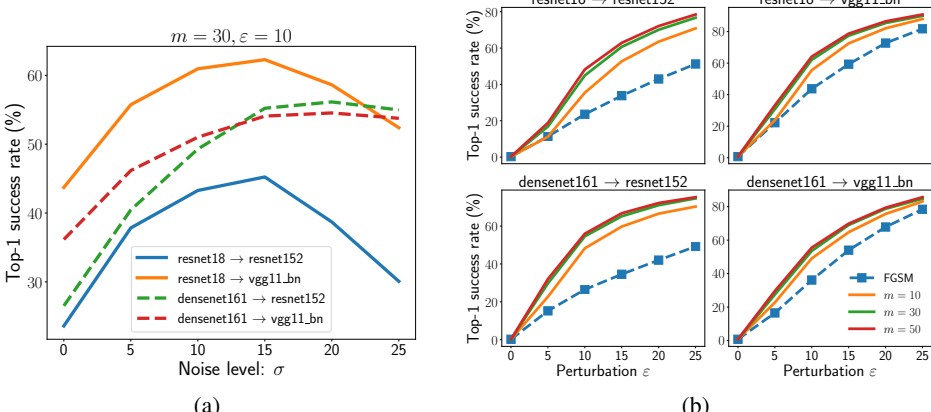

(a)  (b)

Figure 3: **(a)** The sensitivity of the hyper parameter $\sigma$. **(b)** Success rates for nr-FGSM attacks with different $m$.

## 6.2 ROBUSTNESS ANALYSIS OF ADVERSARIAL EXAMPLES

In this part, we preliminarily explore the robustness of adversarial perturbations to image transformations, such as rotation, scaling, blurring, etc. This property is particularly important in practice, since it directly affects whether adversarial examples can survive in the physical world (Kurakin et al. (2016); Athalye & Sutskever (2017); Lu et al. (2017)). To quantify the influence of transformations, we use the notion of destruction rate defined in Kurakin et al. (2016),

$$R = \frac{\sum_{i=1}^{N} c(\boldsymbol{x}_i)\left(1 - c(\boldsymbol{x}_i^{adv})\right) c(T(\boldsymbol{x}_i^{adv}))}{\sum_{i=1}^{N} c(\boldsymbol{x}_i)\left(1 - c(\boldsymbol{x}_i^{adv})\right)},$$

where $N$ is the number of images used to estimate the destruction rate, $T(\cdot)$ is an arbitrary image transformation. The function $c(\boldsymbol{x})$ indicates whether $\boldsymbol{x}$ is classified correctly:

$$c(\boldsymbol{x}) := \begin{cases} 1, & \text{if image } \boldsymbol{x} \text{ is classified correctly} \\ 0, & \text{otherwise} \end{cases}$$

And thus, the above rate $R$ describes the fraction of adversarial images that are no longer misclassified after the transformation $T(\cdot)$.

*Densenet121* and *resnet34* are chosen as our source and target model, respectively; and four image transformations are considered: rotation, Gaussian noise, Gaussian blur and JPEG compression. Figure 4 displays the results, which show that adversarial examples generated by our proposed NRG methods are much more robust than those generated by vanilla methods.

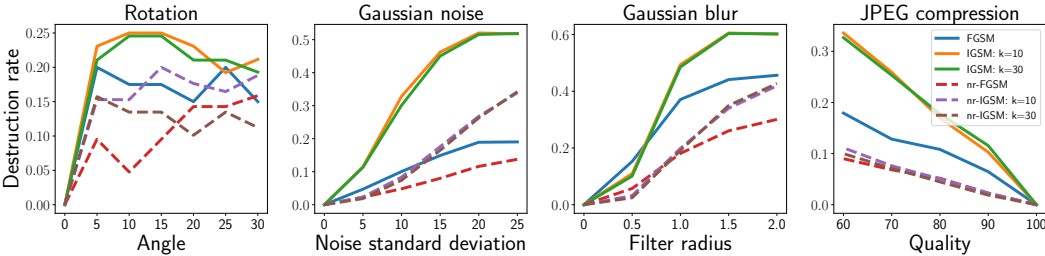

Figure 4: Destruction rates of adversarial examples for various methods. For NRG methods, we choose $m = 20, \sigma = 15$. The distortion $\varepsilon$ is set to 15.

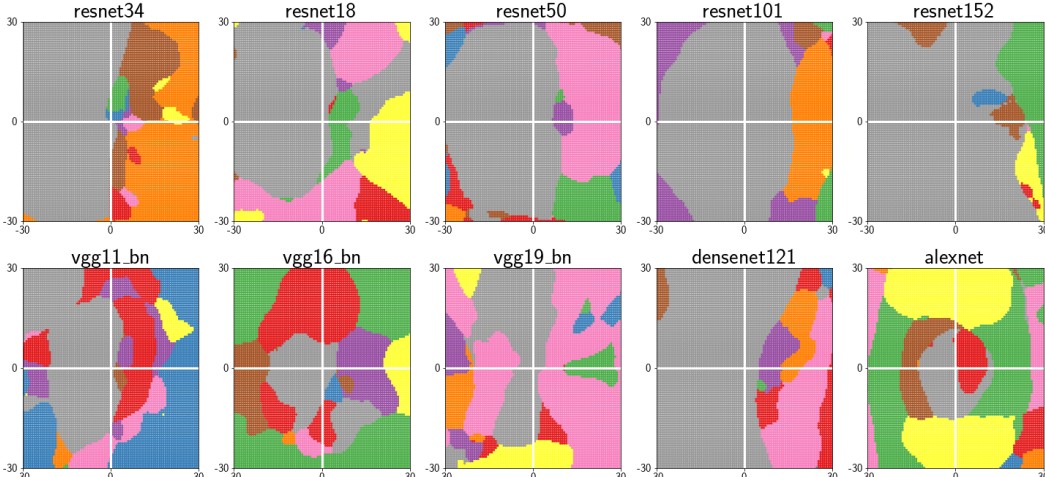

Figure 5: Decision boundaries of various models. The first one is the source model *resnet34* and the others are the target models examined. The horizontal axis represents $\text{sign}\left(\nabla f_{\parallel}\right)$ of *resnet34*, and the vertical axis indicates $\text{sign}\left(\nabla f_{\perp}\right)$. For each figure, the same color indicates the same predicted label. The origin represents the clean image $x$, whose label is *table lamp* with ID in ILSVRC2012 validation set being 26 ( shown in Figure 8 of Appendix C ).

### 6.3 VISUALIZATION OF DECISION BOUNDARY

In this section we study the decision boundaries of different models to help us understand why NRG-based methods perform better. *Resnet34* is chosen as the source model and nine target models are considered, including *resnet18, resnet50, resnet101, resnet152, vgg11_bn, vgg16_bn, vgg19_bn, densenet121, alexnet*. The $\nabla f_{\parallel}$ is estimated by (8) with $m = 1000, \sigma = 15$. Each point $(u, v)$ in this 2-D plane corresponds to the image perturbed by $u$ and $v$ along $\text{sign}\left(\nabla f_{\parallel}\right)$ and $\text{sign}\left(\nabla f_{\perp}\right)$, respectively, i.e.

$$\text{clip}\left(x + u\,\text{sign}\left(\nabla f_{\parallel}\right) + v\,\text{sign}\left(\nabla f_{\perp}\right), 0, 255\right) \tag{13}$$

where $x$ represents the original image. We randomly select one image that can be recognized by all the models examined. The Figure 5 shows the decision boundaries. We also tried a variety of other source models and images, all the plots are similar.

From the aspect of changing the predicted label, the direction of $\text{sign}\left(\nabla f_{\parallel}\right)$ is as sensitive as the direction of $\text{sign}\left(\nabla f_{\perp}\right)$ for the source model *resnet34*. However, except *alexnet* all the other target models are much more sensitive along $\text{sign}\left(\nabla f_{\parallel}\right)$ than $\text{sign}\left(\nabla f_{\perp}\right)$. This is consistent with the argument in Section 3.1. Removing $\nabla f_{\perp}$ from gradients can also be thought as penalizing the optimizer along the model-specific direction, thus avoiding converging to a source model-overfitting solution that transfers poorly to the other target models.

Moreover, we find that, along the $\text{sign}\left(\nabla f_{\parallel}\right)$, the minimal distance $u$ to produce adversarial transfer varies for different models. The distances for complex models are significantly larger than those of small models, for instance, the comparison between *resnet152* and *resnent50*. This provides us a geometric understanding of why big models are more robust than small models, as observed in Section 5.1.

### 6.4 INFLUENCE OF MODEL CAPACITY AND ACCURACY ON ATTACK CAPABILITY

In earlier experiments (Table 1 and Table 2), we can observe that adversarial examples crafted from *alexnet* generalize worst across models, for example nr-FGSM attack of *alexnet* → *resnet152* only achieves 19.3 percent. However, attacks from *densenet121* consistently perform well for any target model, for example $84.3\%$ of nr-IGSM adversarial examples can transfer to *vgg19_bn*, whose architecture is completely different from *densenet121*. This observation indicates that different models can exhibit different performances in attacking the same target model. Now we attempt to find the

principle behind this important phenomenon, which can guide us to choose a better local model to generate adversarial examples for attacking the remote black-box system.

We select *vgg19_bn* and *resnet152* as our target model, and use a variety of models to perform both FGSM and IGSM attacks against them. The results are summarized in Figure 6. The horizontal axis is the Top-1 test error, while the vertical axis is the number of model parameters that roughly quantifies the model capacity. We can see that the models with powerful attack capability concentrate in the bottom left corner, while fooling rates are very low for those models with large test error and number of parameters[3]. We can obtain an important observation that

> *the smaller test error and lower capacity a model has, the stronger its attack capability is.*

Here we attempt to explain this phenomenon from our understanding of transferability in Section 3.1. A smaller test error indicates a lower bias for approximating the ground truth along the data manifold. On the other hand, a less complex model might lead to a smaller model-specific component $\nabla f_\perp$, facilitating the data-dependent factor dominate. In a nutshell, the model with small $\nabla f_\perp$ and large $\nabla f_\parallel$ can provide strong adversarial examples that transfer more easily. This is consistent with our arguments presented previously.

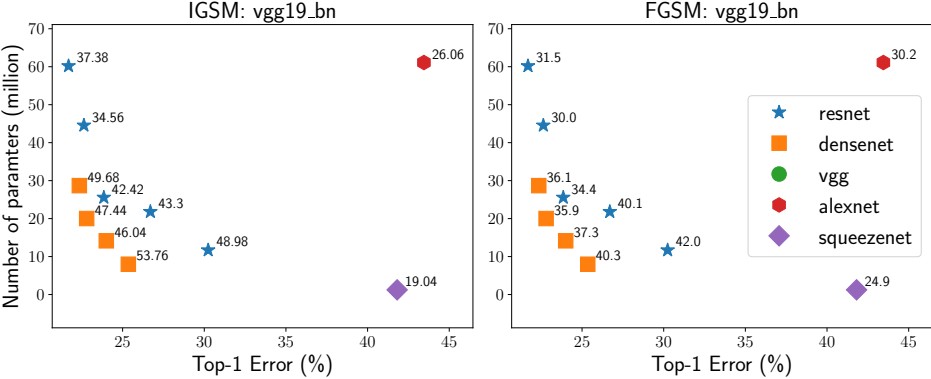

Figure 6: Top-1 success rates of FGSM and IGSM ($k = 20, \alpha = 5$) attacks against *vgg19_bn* for various models. The annotated value is the percentage of adversarial examples that can transfer to the *vgg_19*. Here, the models of vgg-style have been removed, since the contribution from architecture similarities is not in our consideration. The distortion is chosen as $\varepsilon = 15$. The plots of attacking *resnet152* are similar and can be found in Appendix C (Figure 9).

## 7 CONCLUSION

In this paper, we have verified that an adversarial perturbation can be decomposed into two components: model-specific and data-dependent ones. And it is the latter that mainly contributes to the transferability of adversarial examples. Based on this understanding, we proposed the noise-reduced gradient (NRG) based methods to craft adversarial examples, which are much more effective than previous methods. We also show that the models with lower capacity and higher test accuracy are endowed with stronger capability for black-box attacks.

In the future, we will consider combining NRG-based methods with adversarial training to defend against black-box attacks. The component contributing to the transferability is data-dependent, which is intrinsically low-dimensional, so we hypothesize that black-box attacks can be defensible. On the contrary, the white-box attack origins from the extremely high-dimensional ambient

---

[3]Note that the number of model parameters is only an approximate measure of model capacity. For the models with the same number of parameters but different architectures, their capacities might be different. To further investigate the relationship between capacity and attack capability, we fix one particular architecture, e.g. fully connected networks for MNIST, and compare this architecture with different layers, as shown in Appendix B. And the observation is still consistent with that on ImageNet data.

space, thus its defense is much more difficult. Another interesting thread of future research is to learn stable features beneficial for transfer learning by incorporating our NRG strategy, since the reduction of model-specific noise can lead to more accurate information on the data manifold.

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

APPENDIX A: INFLUENCE OF STEP SIZE FOR TARGETED ATTACKS

When using IGSM to perform targeted black-box attacks, there are two hyper parameters including number of iteration $k$ and step size $\alpha$. Here we explore their influence to the quality of adversarial examples generated. The success rates are valuated are calculated on $1,000$ images randomly selected according to description of Section 4. *resnet152* and *vgg16_bn* are chosen as target models. The performance are evaluated by the average Top-5 success rate over the three ensembles used in Table 3(b).

Figure 7 shows that for the optimal step size $\alpha$ is very large, for instance in this experiment it is about 15 compared to the allowed distortion $\varepsilon = 20$. Both too large and too small step size will yield harm to the attack performances. It interesting noting that with small step size $\alpha = 5$, the large number of iteration provides worser performance than small number of iteration. One possible explain is that more iterations lead optimizers to converge to more overfit solution. In contrast, a large step size can prevent it and encourage the optimizer to explore more model-independent area, thus more iteration is better.

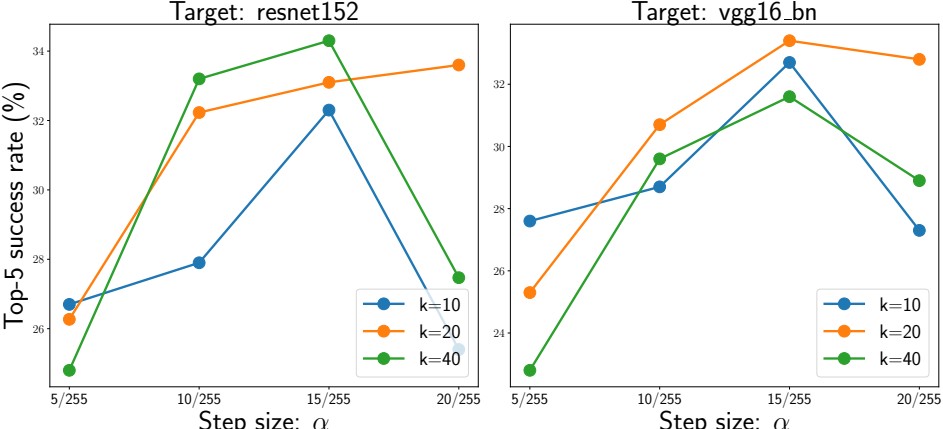

Figure 7: Average success rates over three ensembles for different step size $\alpha$ and number of iteration $k$. The three ensembles are the same with those in Table 3(b). Distortion $\varepsilon = 20$.

APPENDIX B: MNIST EXPERIMENTS

To further confirm the influence of model redundancy on the attack capability, we conduct an additional experiment on MNIST dataset. We use the fully networks of $D$ layers with width of each layer being 500, e.g. the architecture of model with $D = 2$ is $784 - 500 - 500 - 10$. Models of depth $1, 3, 5, 9$ are considered in this experiment. The Top-1 success rates of cross-model attacks are reported in Table 4.

The results of Table 4 demonstrate the low-capacity model has much stronger attack capability than large-capacity. This is consistent with our observation in Section 6.4.

Table 4: Top-1 success rates of FGSM attacks with $\varepsilon = 40$. Each cell $(S,T)$ indicates the percentage of adversarial examples generated for model $S$ evaluated on model $T$.

|         | $D = 1$ | $D = 3$ | $D = 5$ | $D = 9$ |
|---------|---------|---------|---------|---------|
| $D = 1$ | 89.37   | 62.94   | 62.90   | 64.42   |
| $D = 3$ | 52.86   | 60.32   | 48.25   | 49.38   |
| $D = 5$ | 47.30   | 43.05   | 55.84   | 44.83   |
| $D = 9$ | 31.19   | 29.22   | 29.02   | 39.60   |

APPENDIX C: SOME ADDITIONAL EXPERIMENTAL RESULTS

Table 5: Top-1 Success rates of non-targeted FGM and vr-FGM $\ell_2$-attacks between pairs of models. The cell (S,T) indicates the percentage of adversarial examples generated for model S(row) that successfully fool model T(column). For each cell: (1) The left is the success rate of FGM; (2) The right rate is the one of vr-FGM. In this experiment, distortion $\varepsilon = 15$.

|  | alexnet | densenet121 | resnet152 | resnet34 | vgg13_bn | vgg19_bn |
|---|---|---|---|---|---|---|
| alexnet | 98.0 / 97.0 | 27.2 / 30.5 | 19.9 / 24.6 | 38.1 / 41.0 | 47.2 / 46.2 | 34.5 / 39.0 |
| densenet121 | 53.5 / 61.6 | 93.1 / 98.6 | 34.5 / 68.7 | 43.6 / 76.2 | 50.7 / 72.8 | 44.2 / 74.6 |
| resnet152 | 50.2 / 54.1 | 37.9 / 67.9 | 82.8 / 95.7 | 45.5 / 72.5 | 40.6 / 63.4 | 35.7 / 64.7 |
| resnet34 | 55.1 / 61.3 | 45.8 / 72.2 | 36.6 / 68.3 | 94.5 / 98.5 | 50.7 / 70.6 | 45.2 / 70.6 |
| vgg13_bn | 45.4 / 55.1 | 25.7 / 53.8 | 14.9 / 38.1 | 28.6 / 59.3 | 97.0 / 98.3 | 53.7 / 86.6 |
| vgg19_bn | 42.9 / 53.3 | 27.4 / 64.0 | 18.1 / 48.4 | 32.0 / 63.2 | 57.1 / 85.9 | 91.8 / 97.7 |

Table 6: Top-1 success rates of ensemble-based non-targeted FGSM and nr-FGSM attacks. Each cell (S,T) indicate the percentages of targeted adversarial examples are predicted as the target label by model (T). The left is the result of FGSM, while the right is the ones of nr-FGSM. Distortion $\varepsilon = 15$

| Ensemble | densenet121 | resnet152 | resnet50 | vgg13_bn |
|---|---|---|---|---|
| resnet18+resnet34+resnet101 | 62.0 / **86.0** | 59.0 / **86.2** | 67.3 / **89.6** | 63.7 / 83.2 |
| vgg11_bn+densenet161 | 52.3 / 81.9 | 35.8 / 70.5 | 48.0 / 78.8 | 69.3 / 89.6 |
| resnet34+vgg16_bn+alexnet | 50.8 / 80.2 | 38.3 / 73.3 | 52.8 / 82.3 | 74.8 / **92.3** |

Table 7: Top-1 success rates of ensemble-based non-targeted IGSM and nr-IGSM attacks. Each cell (S,T) indicate the percentages of targeted adversarial examples are predicted as the target label by model (T). The left is the result of IGSM ($k = 100, \alpha = 3$), while the right is the ones of nr-IGSM ($k = 50, \alpha = 3, m = 20, \sigma = 15$). Since targeted attacks are much more difficult, we choose $\varepsilon = 20$.

| Ensemble | densenet121 | resnet152 | resnet50 | vgg13_bn |
|---|---|---|---|---|
| resnet18+resnet34+resnet101 | 87.8 / **97.8** | 94.6 / **98.9** | 97.4 / **99.4** | 84.1/96.1 |
| vgg11_bn+densenet161 | 86.8 / 97.2 | 62.9 / 89.7 | 80.3 / 94.8 | 94.9 / 98.4 |
| resnet34+vgg16_bn+alexnet | 68.9 / 91.3 | 54.6 / 87.2 | 77.9 / 96.2 | 98.1 / **99.1** |

Table 8: Top-1 success rates of ensemble-based targeted IGSM and nr-IGSM attacks. The cell (S,T) indicates the percentages of targeted adversarial examples generated from model S(row) are predicted as the target label by model T(column). For each cell: The left is the results of IGSM ($k = 20, \alpha = 15$), while the right is the ones of nr-IGSM ($k = 20, \alpha = 15, m = 20, \sigma = 15$). Since targeted attacks are harder, we set $\varepsilon = 20$.

| Ensemble | resnet152 | resnet50 | vgg13_bn | vgg16_bn |
|---|---|---|---|---|
| resnet101+densenet121 | 11.6 / 37.1 | 11.9 / 34.5 | 2.6 / 10.5 | 2.6 / 14.1 |
| resnet18+resnet34+resnet101+densenet121 | 30.3 / **55.2** | 36.8 / **57.3** | 10.8 /**29.1** | 12.8/35.0 |
| vgg11_bn+vgg13_bn+resnet18+ resnet34+densenet121 | 10.1 / 35.1 | 22.2 / 47.9 | - | 42.1/**72.1** |

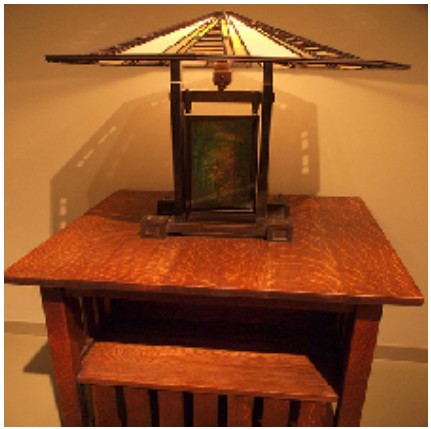

Figure 8: The image used to perform decision boundary analysis. Its ID in ILSVRC2012 validation set is 26, with ground truth label being *table lamp*.

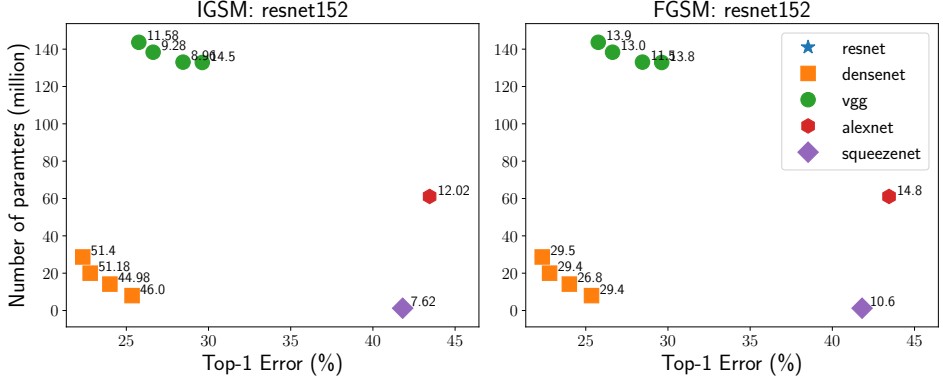

Figure 9: Top-1 success rates of FGSM and IGSM ($k = 20, \alpha = 5$) attacks against *resnet152* for various models. The annotated value is the percentage of adversarial examples that can transfer to the *resnet152*. Here, the models of resnet-style have been removed, since the contribution from architecture similarities is not in our consideration. The distortion is chosen as $\varepsilon = 15$.

