# OpenReview forum: "Enhancing the Transferability of Adversarial Examples with Noise Reduced Gradient"
_ICLR.cc/2018/Conference — Reject_

### Official Review · AnonReviewer2 · 2017-11-24

**Rating:** 4
**Confidence:** 4

**Review:**

This paper postulates that an adversarial perturbation consists of a model-specific and data-specific component, and that amplification of the latter is best suited for adversarial attacks.

This paper has many grammatical errors. The article is almost always missing from nouns. Some of the sentences need changing. For example:

"training model paramater"  --> "training model parameters" (assuming the neural networks have more than 1 parameter)
"same or similar dataset with" --> "same or a similar dataset to"
"human eyes" --> "the human eye"!
"in analogous to" --> "analogous to"
"start-of-the-art" --> "state-of-the-art"

Some roughly chronological comments follow:

In equation (1) although it is obvious that y is the output of f, you should define it. As you are considering the single highest-scoring class, there should probably be an argmax somewhere.

"The best metric should be human eyes, which is unfortunately difficult to quantify". I don't recommend that you quantify things in terms of eyes.

In section 3.1 I am not convinced there is yet sufficient justification to claim that grad(f||)^A is aligned with the inter-class deviation. It would be helpful to put equation (8) here. The "human" line on figure 1a doesn't make much sense. By u & v in the figure 1 caption you presumably the x and y axes on the plot. These should be labelled.

In section 4 you write "it is meaningless to construct adversarial perturbations for the images that target models cannot classify correctly". I'm not sure this is true. Imagenet has a *lot* of dog breeds. For an adversarial attack, it may be advantageous to change the classification from "wrong breed of dog" to "not a dog at all".

Something that concerns me is that, although your methods produce good results, it looks like the hyperparameters are chosen so as to overfit to the data (please do correct me if this is not the case). A better procedure would be to split the imagenet validation set in two and optimise the hyperparameters on one split, and test on the second. You also "try lots of \alphas", which again seems like overfitting.

Target attack experiments are missing from 5.1, in 5.2 you write that it is a harder problem so it is omitted. I would argue it is still worth presenting these results even if they are less flattering.

Section 6.2 feels out of place and disjointed from the narrative of the paper.

A lot of choices in Section 6 feel arbitrary. In 6.3, why is resnet34 the chosen source model? In 6.4 why do you select those two target models?

I think this paper contains an interesting idea, but suffers from poor writing and unprincipled experimentation. I therefore recommend it be rejected.

Pros:
- Promising results
- Good summary of adversarial methods

Cons:
-  Poorly written
-  Appears to overfit to the test data

---

### Official Review · AnonReviewer1 · 2017-11-26
**Some arguments are not well justified**

**Rating:** 5
**Confidence:** 3

**Review:**

This paper focuses on enhancing the transferability of adversarial examples from one model to another model. The main contribution of this paper is to factorize the adversarial perturbation direction into model-specific and data-dependent. Motivated by finding the data-dependent direction, the paper proposes the noise reduced gradient method.

The paper is not mature. The authors need to justify their arguments in a more rigorous way, like why data-dependent direction can be obtained by averaging; is it true factorization of the perturbation direction? i.e. is the orthogonal direction is indeed model-specific? most of explanations are not rigorous and kind of superficial.

---

### Official Review · AnonReviewer3 · 2017-11-29
**Interesting study of the most intriguing but lesser studied aspect of adversarial examples.**

**Rating:** 5
**Confidence:** 4

**Review:**

The problem of exploring the cross-model (and cross-dataset) generalization of adversarial examples is relatively neglected topic. However the paper's list of related work on that toopic is a bit lacking as in section 3.1 it omits referencing the "Explaining and Harnessing..." paper by Goodfellow et al., which presented the first convincing attempt at explaining cross-model generalization of the examples.

However this paper seems to extend the explanation by a more principled study of the cross-model generalization. Again Section 3.1. presents a hypothesis on splitting the space of adversarial perturbations into two sub-manifolds. However this hypothesis seems as a tautology as the splitting is engineered in a way to formally describe the informal statement. Anyways, the paper introduces a useful terminology to aid analysis and engineer examples with improved generalization across models.

In the same vain, Section 3.2 presents another hypothesis, but is claimed as fact. It claims that the model-dependent component of adversarial examples is dominated by images with high-frequency noise. This is a relatively unfounded statement, not backed up by any qualitative or quantitative evidence.

Motivated by the observation that most newly generated adversarial examples are perturbations by a high frequency noise and that noise is often model-specific (which is not measured or studied sufficiently in the paper), the paper suggests adding a noise term to the FGS and IGSM methods and give extensive experimental evidence on a variety of models on ImageNet demonstrating that the transferability of the newly generated examples is improved.

I am on the fence with this paper. It certainly studies an important,  somewhat neglected aspect of adversarial examples, but mostly speculatively and the experimental results study the resulting algorithm rather than trying trying the verify the hypotheses on which those algorithms are based upon.

On the plus side the paper presents very strong practical evidence that the transferability of the examples can be enhanced by such a simple methodology significantly.

I think the paper would be much more compelling (are should be accepted) if it contained a more disciplined study on the hypotheses on which the methodology is based upon.

---

### Decision · Program_Chairs · 2018-01-29
**ICLR 2018 Conference Acceptance Decision**

**Decision:**

Reject

**Comment:**

The paper studies transferability of adversarial examples between model architectures, and proposes a method to improve this transferability. Whilst it covers an interesting and relevant line of research, the paper does not provide strong evidence for its main underling hypothesis: namely, that adversarial perturbations can be split in a model-specified and a data-specific part. The paper's presentation also warrants improvements. The authors have not provided a rebuttal.